# Accurate Diagnosis of High-Risk Pulmonary Nodules Using a Non-Invasive Epigenetic Biomarker Test

**DOI:** 10.3390/cancers17060916

**Published:** 2025-03-07

**Authors:** Pei-Hsing Chen, Tung-Ming Tsai, Tzu-Pin Lu, Hsiao-Hung Lu, Dorian Pamart, Aristotelis Kotronoulas, Marielle Herzog, Jacob Vincent Micallef, Hsao-Hsun Hsu, Jin-Shing Chen

**Affiliations:** 1Institute of Biomedical Engineering, College of Medicine and College of Engineering, National Taiwan University, Taipei City 106, Taiwan; chenph@ntu.edu.tw; 2Department of Surgery, National Taiwan University Hospital and National Taiwan University College of Medicine, Taipei City 100, Taiwan; tplu@ntu.edu.tw (T.-P.L.); ntuhsu@gmail.com (H.-H.H.); 3Department of Surgical Oncology, National Taiwan University Cancer Center, College of Medicine, National Taiwan University, Taipei City 106, Taiwan; 4Institute of Epidemiology and Preventive Medicine, College of Public Health, National Taiwan University, Taipei City 100, Taiwan; 5Belgian Volition SRL, 22 Rue Phocas Lejeune, Parc Scientifique Crealys, 5032 Isnes, Belgium; d.pamart@volition.com (D.P.); m.herzog@volition.com (M.H.); 6Volition Diagnostics UK Limited, 93-95 Gloucester Place, London W1U 6JQ, UK

**Keywords:** lung cancer, epigenetic biomarkers, histone PTMs, circulating nucleosomes, pulmonary nodule, liquid biopsy

## Abstract

The early detection of lung cancer is crucial for improving patient outcomes, but current methods are limited. This research aims to enhance the identification of malignant lung nodules using a blood test that assesses nucleosome levels and histone modifications. By analyzing blood samples from over 800 patients with lung nodules larger than 5 mm, the researchers developed a model to distinguish between malignant and benign nodules. The model showed high accuracy, particularly in smaller nodules, and was effective in a high-risk patient group. This new method could provide a reliable, non-invasive alternative for detecting lung cancer at an early stage, offering a promising tool for clinical use and improving patient care.

## 1. Introduction

Lung cancer remains the leading cause of cancer-related mortality worldwide [1]. Early detection using low-dose computed tomography (LDCT) reduces death rates, as evidenced in pivotal trials such as the National Lung Screening Trial (NLST) [2] and the Dutch-Belgian Lung Cancer Screening Trial (NELSON) [3]. The NLST reported a 20% reduction in lung cancer mortality with the implementation of LDCT screening [2]. Despite these outcomes, indeterminate pulmonary nodules, which comprise 50–76% of the nodules identified in the LDCT, present a significant challenge [4,5]. The likelihood of malignancy increases with nodule size, with those measuring 7–29 mm exhibiting a malignancy risk ranging from 1.7% to 22% [6]. Lung nodules detected using LDCT screening are often <20 mm, complicating the biopsy process [6,7,8]. Consequently, while close monitoring is the primary strategy, larger tumors may develop acquired or primary resistance or metastasize during observation.

Efforts have been made to develop robust, sensitive, and non-invasive tests to diagnose pulmonary nodules [9]. Despite advancements, there is currently no regulatory-body-approved and widely adopted blood test for the early detection of lung cancer [4]. Tumor cells release various biomolecules such as cell-free DNA (cfDNA), circulating tumor DNA (ctDNA), exosomes, microRNAs, circular RNAs, circulating tumor cells (CTCs), and DNA-methylated fragments. However, while these biomarkers are effective diagnostic biomarkers, some experts remain skeptical of liquid biopsies, particularly in early-stage cancers. Additionally, false positives and false negatives remain a challenge, as some circulating tumor markers may be present without active malignancy, while small tumors may not shed enough detectable biomarkers.

The detection of molecular changes in evolving tumor cells requires highly sensitive and specific assays for ctDNA mutations [10,11]. The diagnostic sensitivity of liquid biopsy tests is hampered by very low levels of somatic molecular alterations in patients with early-stage cancer, who constitute the majority of the population after LDCT screening [12,13].

Changes in DNA methylation in specific regions, such as promoter CpG islands, may signify early molecular events in tumor initiation [14]. Patients with cancer have distinct histone post-translational modifications (PTMs) in circulating nucleosomes, indicating their potential as cancer biomarkers. PTMs regulate chromatin-mediated gene expression, affecting processes such as inflammation, cell cycle, apoptosis, and tumor suppression in lung cancer. Nucleosomes are stable circulating nucleoprotein complexes carrying cfDNA and ctDNA. Unlike DNA analysis, which requires additional preparation and sequencing or PCR, nucleosomes can be directly quantified from plasma via immunoassay. Their measurement is fast, automatable, and suitable for clinical practice. Histone variants and modifications have shown prognostic significance in various cancers. However, their potential as biomarkers for lung nodule differentiation remains underexplored [15,16].

We aimed to develop an epigenetic biomarker (EB) model based on circulating nucleosomes, including histone variant and histone methylation, to evaluate the risk of malignancy in pulmonary nodules and to achieve a more accurate classification of pulmonary nodules, particularly through focusing on identifying malignant nodules in thoracic surgery scenarios.

## 2. Materials and Methods

### 2.1. Study Design and Patients

This prospective blood specimen collection and retrospective evaluation study was approved by the Institutional Review Boards (201905009RIFC) of the participating hospitals and researchregistry-10711. In guidelines for Asia, it is recommended that nodules with a diameter of ≥5 mm undergo clinical management [17]. We recruited 806 participants with undiagnosed nodules larger than 5 mm, identified on computed tomography (CT) scans and classified as high risk by the attending physician. Adult patients of either sex aged ≥18 years were eligible for inclusion if they met the following criteria: pulmonary nodules > 5 mm detected using standard CT or LDCT screening and with nodules categorized as solid nodules, part-solid nodules (mixed ground-glass nodules), or pure non-solid nodules. Participants were recruited from the outpatient clinics of the National Taiwan University Hospital and the National Taiwan University Cancer Center, both teaching hospitals. The study was conducted from August 2019 to July 2021. Exclusion criteria included patients exhibiting metastatic symptoms such as pleural effusion, patients unwilling to undergo blood sampling, patients without a confirmed pathological diagnosis post-surgery, or patients with cancer confirmed pathologically within two years prior to enrollment. Written informed consent was obtained from the patient for blood sampling.

#### 2.1.1. Blood Sampling

All blood samples were prospectively collected before surgery, either during the initial nodule check or during the admission period. Blood (10 mL) was collected in K2-EDTA blood tubes (Sarstedt, Nümbrecht, Germany) within two weeks prior to the initiation of surgery. Blood collection and CT-based response evaluations were conducted for patients undergoing observation at 1–2 weeks intervals. Blood samples were centrifuged at 3000× *g* for 10 min at 15–30 °C. The plasma was then stored at −80 °C until the nucleosome analysis was conducted.

#### 2.1.2. Quantification of Circulating Nucleosomes Using Immunoassays

All samples were tested using Nu.Q^®^ assays (Belgian Volition SRL, Isnes, Belgium). Two nucleosome structures were measured using the Nu.Q^®^ H3.1 and Nu.Q^®^ H3K27Me3 immunoassays according to the manufacturer’s instructions. These sandwich immunoassays, based on chemiluminescence technology, were performed using the IDS-i10 automated analyzer system (Immunodiagnostic System Ltd., Boldon, UK) with a wavelength range of 300 to 500 nm. Briefly, plasma samples were centrifuged at high speed for 2 min and 50 µL of K2-EDTA plasma was incubated with acridinium ester labeled anti-nucleosome detection antibody. Magnetic particle beads coated with the corresponding monoclonal anti-histone variant H3.1 or anti-histone modification H3K27Me3 capture antibody were added. After washing, trigger solutions were added, and the light emitted by the acridinium ester was measured using a luminometer. The results were expressed in relative light units, and concentration (expressed in ng/mL) was extrapolated using four-parameter logistic regression of a reference standard curve. All samples were analyzed in duplicate. If the sample concentration was higher than the lowest concentration and the percent coefficient of variation (%CV) of the determined concentration was >20%, the analysis was repeated.

#### 2.1.3. Chest CT Imaging and Radiological Analysis

For each participant who underwent a chest CT scan (General Electric, Boston, MA, USA/Siemens Healthineers, Erlangen, Germany), the lung and mediastinal image series were reconstructed with a slice thickness of 1.000–1.250 mm if the scans were performed at the National Taiwan University Hospital and National Taiwan University Cancer Center. The slice thickness was at least 5.000 mm. The chest CT images were initially evaluated before drawing blood to confirm inclusion. The diagnosis of the nodules was based on the pathological outcomes. If a nodule disappeared on subsequent imaging and a radiologist confirmed this, it was classified as benign.

Imaging analysis was overseen by Jin-Shing Chen, a senior thoracic surgeon with extensive experience exceeding three decades. Additional team members, Pei-Hsing Chen and Tung-Ming Tsai, with 10 and 17 years of experience, respectively, interpreted the scans and delineated the regions of interest. For subgroup analysis, the lung imaging, reporting, and data system (Lung-RADS) version 1.1 guidelines were used [18]. In the validation of the model, Hsiao-Hung Lu performed a blinded assessment, categorizing the nodules as malignant or benign based on their spiculation characteristics.

Tumor size was determined preoperatively based on thin-section CT findings. All tumors were subsequently evaluated to estimate the extent of ground-glass opacity (GGO) using a thin-section CT scan with a 5.000 mm collimation. The solid component, part-solid component, and GGO were defined as areas of increased opacification that completely obscured the underlying vascular markings, as described in previous studies [19,20]. GGO was defined as an area of slight, homogeneous increase in density that did not obscure the underlying vascular markings.

#### 2.1.4. Operation Policy and Pathology Evaluation

The operation policy followed the American College of Chest Physicians or American Association for Thoracic Surgery recommendations [21] to evaluate and treat nodules > 5 mm [17]. Malignant and benign tumors were defined based on the 2021 WHO Classification of Lung Tumors [22]. The subcategories of adenocarcinoma included in the malignant group were atypical adenomatous hyperplasia (AAH), adenocarcinoma in situ, minimally invasive adenocarcinoma, and invasive adenocarcinoma [23].

### 2.2. Theory/Calculation

#### 2.2.1. EB Model Development for Benign–Malignant Predictions

To advance the prediction of benign and malignant states in lung cancer, we developed an EB model specifically focusing on nucleosome levels and histone modifications in circulating blood. For model building, we employed a logistic regression approach to predict benign and malignant states. For model development, we randomly allocated 25% of the data to the test dataset (n = 202). The remaining 75% were further divided, with 80% (n = 483) used for training and 20% (n = 121) for validation (Figure 1).

The training dataset was used for model development and the calculation of its coefficients. Key biomarkers included histone isoform nucleosome levels (Nu.Q^®^ H3.1) and methylated lysine 27 of histone H3 (Nu.Q H3K27Me3). A validation dataset was used to fine-tune the model and ensure its predictive capability.

Given the benefits of video-assisted thoracoscopic surgery, such as reduced invasiveness, accurate localization, and quick recovery [21,24,25], the main challenge for thoracic surgeons is to accurately identify malignant lung nodules and minimize the risk of delayed diagnosis in high-risk populations identified through CT/LDCT screening. This necessitates decreasing the false-negative rate while maintaining an adequate positive predictive value (PPV). Achieving this requires a sensitivity of >80% while maintaining an adequately high PPV to confidently identify lung cancer in high-prevalence populations. To this end, the probability cutoff for cancer diagnosis was determined by maintaining the model’s sensitivity at >0.80 and then selecting the optimal threshold using the Youden index. Finally, this cutoff was applied to the test dataset, and the diagnostic performance was calculated.

#### 2.2.2. Comparison with the Mayo Clinic and Veteran Affairs (VA) Models

The Mayo Clinic model for malignancy in pulmonary nodules calculates the probability of malignancy using three clinical and three radiographic variables. The formula is as follows:probability of malignancy = e^x^/(1 + e^x^),
where x = −6.8272 + (0.0391 × age) + (0.7917 × smoking) + (1.3388 × cancer) + (0.1274 × nodule diameter) + (1.0407 × spiculation) + (0.7838 × upper lobe), and e is Euler’s number, a mathematical constant approximately equal to 2.71828 [26].

The VA model for malignancy in pulmonary nodules calculates the probability of malignancy using three clinical variables and one radiographic variable. The formula isprobability of malignancy = 100 × (e^x^/[1 + e^x^]), where x = −8.404 + 2.061 × smoke + 0.779 × age/10 + 0.112 × diameter + 0.567 × yearsquit/10,
where smoking is 1 if the patient is a current or former smoker (otherwise 0); age divided by 10 is the age in years divided by 10; diameter is the largest diameter of the nodule in millimeters; yearsquit/10 is the number of years since quitting smoking divided by 10; and e is Euler’s number [27].

#### 2.2.3. Statistics

All statistical analyses were conducted using R software (version 4.4.1). Categorical variables such as sex and nodule sub-type were compared using Fisher’s exact test. The sensitivities of the different Lung-RADS for malignant nodules were compared using Fisher’s exact test. Continuous variables such as age were compared using Student’s *t*-test, and 95% confidence intervals (CIs) were calculated based on Wald confidence intervals for proportions. A *p*-value < 0.05 was considered statistically significant.

The sensitivity, specificity, accuracy, PPV, and negative PV (NPV) of the model and other models for differentiating malignant nodules were assessed by comparing the pathological outcomes and imaging studies (for those with vanished nodules only). Receiver operating characteristic (ROC) and area under the curve (AUC) were calculated using pROC R package (version 1.15.3) software. The following R packages (version 1.15.3) were utilized: readxl, tidyverse, ggplot2, pROC, tidymodels, and caret.

## 3. Results

### 3.1. Patient Demographics and Clinicopathologic Features

In total, 806 patients who were CT-positive/LDCT-positive were recruited from the thoracic surgery departments of the National Taiwan University Hospital and the National Taiwan University Cancer Center. Of these, 755 (93.7%) were tissue-diagnosed. The remaining 51 (6.3%) patients had nodules that disappeared on subsequent imaging, confirmed by a radiologist, and defined as benign. The malignancy rate in the entire cohort was 80.4% (158 benign, 648 malignant). An overview of the study design is provided in Figure 1, and the demographic characteristics of the 806 patients are detailed in Table 1. The diagnoses in the cohort were predominantly early-stage lung cancer (AAH, stage 0, stage I, or stage II), comprising 84.3% (546/648) of all cancer patients in the cohort. The mean nodule size in the entire cohort was 25 mm. Additionally, 78.6% (630/806) of the patients had never smoked. Adenocarcinoma and its subcategories constituted the majority of malignant cases (92.6%, 600/648). No statistically significant differences in the distribution of malignancy, age, sex, tumor components, tumor size, Lung-RADS scores, or smoking history (*p* > 0.05) were observed among the three datasets. The demographic and clinical characteristics of the participants are presented in Table 1.

### 3.2. The EB Model and Lung Cancer Diagnostic Accuracy

The cohort data were divided into a test dataset (n = 202, 25%) and the EB model development subset (75%). The training dataset (n = 483) comprised 80% of the EB model development subset and was used for the model development. The validation dataset (n = 121) comprised 20% of the model development subset and was used for the optimal threshold selection.

We developed the EB model by screening multiple combination models from five quantitative epigenetic features (Appendix A) derived from blood tests during the pre-training model tuning. For feature selection, we analyzed the relationship between the AUC values and the number of primary features. The AUC plateaued when the number of primary features reached two, indicating that adding more features did not significantly improve the AUC. Therefore, we identified two predictors of malignancy using multivariate logistic regression analysis. This was then applied to the training dataset using Nu.Q^®^ H3.1 and Nu.Q^®^ H3K27Me3. Other potential predictors not associated with malignancy were excluded from the final model. The prediction model was calculated as follows:Probability of malignant SPN = e^x^/1 + e^x^X = 1.78668 + 0.07821 × H3.1 − 0.15885 × H3K27Me3,
where H3.1 is the level of the histone variant H3.1, and H3K27Me3 is the histone modification H3K27Me3. Analysis of the relationship between the AUC values, detailed in Figure 2a,b, provided AUC values for all the datasets. Positive and negative classifications for the model were determined using a cutoff value (0.755). We validated the performance of the EB model using a test set that showed consistently good performance (Appendix A).

When comparing the model performance across all cohorts, specifically for small nodules (5–10 mm), the results indicated that the model remained robust even for smaller nodules. In the overall dataset, the AUCs for the EB model were 0.74, 0.86, and 0.79 for the training, validation, and test datasets, respectively, with accuracies ranging from 0.80 to 0.88. Sensitivity was high across all datasets, with values of 0.91, 0.95, and 0.93, respectively, while specificity ranged from 0.37 to 0.64. The PPV and NPV were consistent, indicating the reliability of the model for predicting true positives and negatives.

The model maintained strong performance for small nodules (5–10 mm). The AUCs of the training, validation, and test datasets were 0.70, 0.89, and 0.80, respectively, with accuracies of 0.76, 0.88, and 0.85, respectively, indicating that the diagnostic accuracy of the model remained high for smaller nodules. The sensitivities were 0.91, 1.00, and 0.94, respectively, indicating that the model correctly identified the majority of malignant cases. Although lower than sensitivity, specificity was sufficient in most of the subgroups to complement high sensitivity, with values of 0.27, 0.62, and 0.54, respectively.

#### Lung-RADS Score Analysis in the Test Dataset

The model’s performance across different RADS categories was evaluated using a test dataset. In RADS 2 (n = 73), the model achieved an AUC of 0.75, an accuracy of 0.84, a sensitivity of 0.90, and a specificity of 0.57. In RADS 3 (n = 19), the AUC was 0.81, with an accuracy of 0.84, a sensitivity of 1.00, and a specificity of 0.40. In RADS 4A + 4B (n = 81), the AUC was 0.78, with an accuracy of 0.82, a sensitivity of 0.92, and a specificity of 0.44. In RADS 4X (n = 29), the model performed best, with an AUC of 0.98, an accuracy of 0.97, a sensitivity of 0.96, and a specificity of 1.00. These results show high diagnostic accuracy, especially in the higher RADS categories, indicating the clinical utility of the model for assessing pulmonary nodules (Table 2).

### 3.3. EB Model Performance in Different Nodule Types

The model demonstrated the detection of lung cancer with accuracy independent of the tumor components; 0.84 for solid and part-solid nodules and 0.86 for GGO nodules in the test dataset. Both GGO and part-solid nodules showed higher PPV when maintaining a similar threshold. The PPVs were 0.86 (95% CI 0.76–0.92) for solid nodules, 0.91 (95% CI 0.76–0.98) for part-solid nodules, and 0.91 (95% CI 0.80–0.97) for GGO nodules. These results highlight the high diagnostic accuracy of the model across different tumor components, indicating its potential utility in the assessment of pulmonary nodules (Table 3).

### 3.4. Conventional Cancer Diagnostic Model Comparison

For the validation (Figure 3 and Appendix A), the EB model achieved an AUC of 0.858 (95% CI 0.779–0.937), significantly outperforming the Mayo Clinic model (AUC, 0.570 [95% CI 0.446–0.694]) and the VA model (AUC, 0.503 [95% CI 0.386–0.621]). The accuracy, sensitivity, and specificity of the EB model are detailed in Table 4. Its superior AUC indicates a higher overall performance compared with the Mayo Clinic and VA models. This finding indicates the potential effectiveness of the EB model in accurately predicting outcomes compared with established clinical models.

## 4. Discussion

Early cancer detection is an effective method for reducing cancer-specific mortality. We analyzed the EB profiles of 806 patients with pulmonary nodules and developed an EB model for pulmonary nodule diagnosis; it showed high sensitivity and accuracy with good PPVs at moderate specificity across various imaging characteristics, nodule types, and stages of lung cancer. The EB model also maintained adequate performance, even for small nodules ranging from 5 to 10 mm, which would help decrease the false-negative rate concerning minimally invasive surgery. In addition, sandwich immuo-based assays are simple and relatively inexpensive. This is the first retrospective study to validate a blood-based EB model for diagnosing lung nodules.

Previous studies have attempted to enhance lung cancer risk assessment using blood-based biomarkers [28,29,30]. Various biosources from liquid biopsy, including cfDNA, ctDNA, CTCs, exosomes, and tumor-educated platelets, have been extensively investigated for their role in lung cancer diagnosis. However, none of these tests have been implemented clinically because their sensitivities and specificities are typically insufficient for clinical decision-making [12,13,30]. Alterations in the epigenome, such as DNA methylation and histone modification, play pivotal roles in carcinogenesis. DNA methylation levels and global histone modification patterns may predict cancer recurrence and prognosis across a wide variety of cancer types [31,32]. In lung cancer, these changes affect significant signaling pathways, including the ERK family, NF-kB, and Hedgehog pathways. Additionally, epigenetic markers are potential biomarkers for early screening, monitoring, and therapeutic strategies in non-small-cell lung cancer (NSCLC) [16].

PTMs can work together or independently to promote the activation or suppression of chromatin-mediated gene expression. They include the regulation of inflammatory cytokines, cell cycle arrest, senescence, apoptosis, growth factors, antioxidants, and tumor suppressor genes associated with lung cancer [16]. We focused on the histone variant H3.1 levels and the histone modification H3K27me3. Regarding the prognostic effect of H3K27me3 in various human cancers, the increased level of H3K27me3 is linked to a more malignant behavior and worse prognosis in patients with prostate [33], esophageal [34], nasopharyngeal [35], and hepatocellular [36] carcinoma. Conversely, in breast, ovarian, and pancreatic cancers [31] and in renal cell carcinoma [37], a decrease in the H3K27me3 levels is associated with a worse prognosis. In lung cancer patients, a lower level of H3K27me3 in tissues has been associated with carcinogenesis [38], whereas a high level of circulating H3K27Me3-nucleosomes in blood has been associated with lung cancer at diagnosis and during treatment [39]. Our model focused on differentiating between benign and malignant nodules. The precise role of H3K27me3 in distinguishing between normal and malignant populations still requires further investigation.

In the era of network medicine and artificial intelligence (AI), integrating epigenetic-sensitive biomarkers like H3K27me3 into clinical practice through patient-centered platforms offers significant potential [40]. These platforms utilize AI to analyze genetic, epigenetic, and clinical data, enabling the precise diagnosis and treatment of lung cancer [41,42]. By processing large-scale omics data, such systems can create personalized treatment plans and provide real-time tumor profile information for tailored interventions. Recent studies [40,41,42] highlight how AI-driven platforms in oncology can improve diagnostic accuracy and predict treatment response. By analyzing circulating H3K27me3-nucleosomes, AI can identify tumor patterns and suggest personalized therapies. This combination of network medicine and AI can enhance cancer detection, monitoring, and treatment selection, ultimately optimizing patient outcomes.

We believe that network medicine and AI will be the future of oncology, increasing diagnostic accuracy and enabling the selection of the right treatment. This includes identifying the highest-risk groups for adjuvant therapy and predicting prognosis based on imaging and pathological risk factors, ensuring more precise and effective personalized care for lung cancer patients.

In the era of network medicine and artificial intelligence (AI), integrating epigenetic-sensitive biomarkers like H3K27me3 into clinical practice through patient-centered platforms offers significant potential. These platforms would utilize AI to analyze genetic, epigenetic, and clinical data, enabling the precise diagnosis and treatment of lung cancer. By processing large-scale omics data, such systems can create personalized treatment plans and provide real-time tumor profile information for tailored interventions. Recent studies highlight how AI-driven platforms in oncology can improve diagnostic accuracy and predict treatment response. By analyzing circulating H3K27me3-nucleosomes, AI can identify tumor patterns and suggest personalized therapies. This combination of network medicine and AI would enhance cancer detection, monitoring, and treatment selection, ultimately optimizing patient outcomes.

Beyond LDCT screening, liquid biopsies can identify various biomolecular markers, offering insights into the disease status. Integrating liquid biopsy with model training shows great promise for early-stage diagnoses [43]. However, many current liquid biopsy methods targeting early cancer detection lack the sensitivity to reliably identify early-stage cancers or small nodules [44,45,46,47]. The EB model showed satisfactory accuracy, PPV, and NPV for small nodules, specifically within the 5–10 mm range (Table 2). This simple epigenetic regression model had both strengths and limitations. The AUC and accuracy of the model improved significantly from the training to the validation dataset, indicating better predictive performance with new data. The sensitivity was particularly strong, with the validation dataset achieving a perfect score (1.000), emphasizing its effectiveness in detecting malignant nodules, which is essential for early cancer detection. However, specificity was sub-optimal, particularly in the training set. While specificity improved in the validation and test datasets, the risk of false positives remained an issue. Given the benefits of video-assisted thoracoscopic surgery, the main challenge for thoracic surgeons is to accurately identify malignant lung nodules and minimize the risk of delayed diagnosis in high-risk populations identified through CT/LDCT screening.

We applied the updated Lung-RADS to retrospectively evaluate nodule characteristics [18]. The Lung-RADS categorizes nodules based on their likelihood of being malignant, with classification depending on characteristics such as size, attenuation, growth pattern, and other features that may indicate a higher risk of cancer [18]. Nodules classified under Lung-RADS categories 1 and 2 have an estimated malignancy risk of <1%, whereas those in category 3 have a 1–2% risk. Nodules in the category 4A have a 5–15% risk, whereas those in categories 4B and 4X have a risk of >15% [18]. Most lung cancers identified through screening were observed in nodules categorized as Lung-RADS 3 or 4.

Table 3 shows that the model effectively maintained good accuracy, PPV, and sensitivity in Lung-RADS categories 2 and 3, which are often associated with a lower malignancy risk. The accuracy of both RADS 2 and RADS 3 was consistently high at 0.84, indicating that the model was reliable for correctly classifying nodules as either malignant or benign within these groups. The PPV remained robust, with RADS 2 at 0.90 and RADS 3 at 0.82, showing that when the model predicted a nodule as malignant, it was generally accurate and that the likelihood that these nodules were malignant was high. Furthermore, the sensitivity findings highlighted the model’s reliability in correctly identifying malignant nodules, ensuring that few malignancies were missed. Overall, the model maintained strong performance across these metrics, demonstrating its reliability and effectiveness even in categories with a lower pre-test probability of malignancy, supporting its utility in lung cancer screening programs.

In CT lung cancer screening, detected nodules are often part-solid or non-solid; these types of nodules are more likely to be malignant than solid nodules, even when their size is considered [48]. Therefore, achieving high accuracy and PPV in non-solid and part-solid nodules is crucial; our model performed very well in these categories. Although advancements in lung cancer treatment now allow surgery even for stage IV patients, early detection remains the best treatment option [49]. As shown in Table 4, the model’s accuracy was consistent across all nodule types, with 0.84 for both solid and part-solid nodules, and slightly higher at 0.86 for non-solid GGO nodules. The sensitivity remained high, particularly for solid nodules at 0.96, followed by 0.91 for non-solid nodules, and 0.89 for part-solid nodules. This indicates that the model effectively correctly identified malignant nodules across these types. Moreover, the PPV was consistently strong, with 0.86 for solid nodules, 0.91 for part-solid nodules, and 0.91 for non-solid nodules. This suggests that when the model predicts a nodule as malignant, there is a high probability that it will be malignant, regardless of the nodule type. The model showed excellent performance in identifying malignancies in both part-solid and non-solid nodules, which are more likely to be malignant than solid nodules. This strong performance in terms of accuracy, sensitivity, and PPV underlines the reliability and effectiveness of the model for lung cancer screening, particularly for nodules that present a higher risk of malignancy. 

This study had some limitations. We only enrolled thoracic department participants with definitive pathological diagnoses, which may limit the generalizability of the findings. Additionally, the imaging sources were not standardized across the studies, leading to potential image quality and interpretation variability. There was also a lack of integration between the imaging and clinical characteristics of the model parameters, owing to the simplified methodology. Such integration would have provided a more comprehensive assessment. Moreover, the study enrolled a higher proportion of women than men, more never-smokers than smokers, and more patients with adenocarcinoma than those with squamous carcinoma. The study was conducted without using a central laboratory in Taiwan and involved a shipping process that may have introduced variability. We plan to establish an EB model test at a central laboratory-provided service in Taiwan to ensure consistency and reliability. Furthermore, this study was conducted retrospectively, which raises the possibility of overfitting during the model development. The lack of validation further limits the validation of the findings’ robustness. A prospective, multi-institutional study with a larger and more diverse patient cohort is required to confirm these observations and enhance the generalizability of the results.

## 5. Conclusions

Our study showed that high sensitivity and accuracy in the early detection of NSCLC can be achieved using a panel of EBs in plasma. Concerning detecting high-risk lung cancer, the EB model performed well in detecting small, part-solid, and non-solid nodules, which are the majority in lung cancer screening. This model may reduce false-negative results and facilitate early diagnosis.

## Figures and Tables

**Figure 1 cancers-17-00916-f001:**
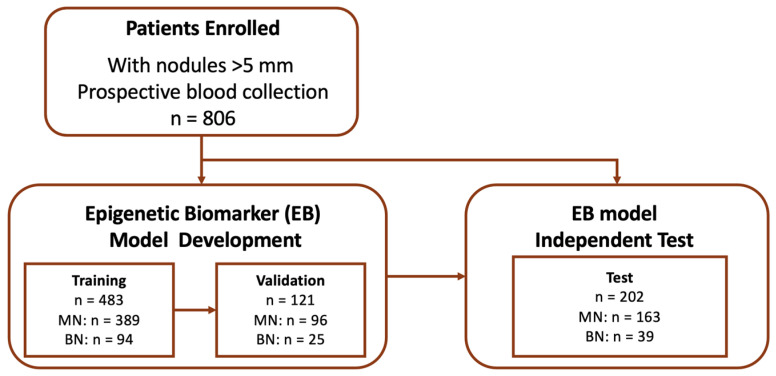
Flowchart of participant enrollment and model development. A total of 806 patients with nodules > 5 mm were enrolled. The Epigenetic Biomarker (EB) model was trained on 483 samples, validated on 121 samples, and tested on 202 samples. The model was built based on the training dataset, with the cutoff determined using the validation dataset, and its performance evaluated on the test dataset. MN refers to malignant nodules, and BN refers to benign nodules.

**Figure 2 cancers-17-00916-f002:**
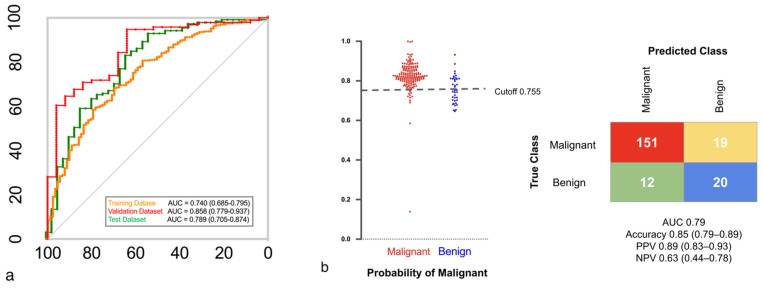
Epigenetic biomarker model. (**a**) A representative ROC curve illustrates the classification performance of the Epigenetic Biomarker model across the training, validation, and test datasets. (**b**) The left panel shows predicted probabilities for malignant (red) and benign (blue) nodules. The right panel presents the confusion matrix of the test dataset at a cutoff value of 0.755, yielding an accuracy of 85%.

**Figure 3 cancers-17-00916-f003:**
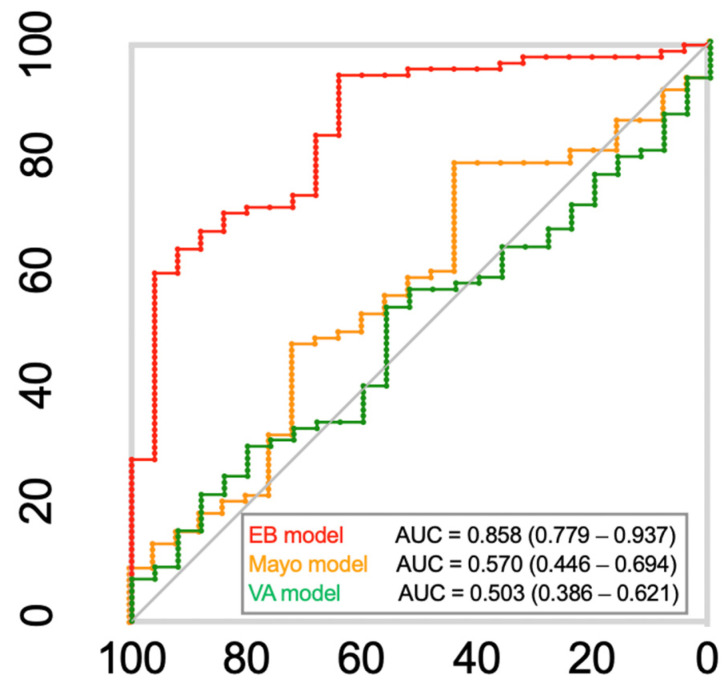
Receiver operating characteristic (ROC) curves for the epigenetic biomarker (EB) model comparing the true class with the predicted class for benign and malignant nodule samples with the Mayo Clinic and Veteran Affairs (VA) models.

**Table 1 cancers-17-00916-t001:** Participants’ baseline characteristics (n = 806).

Patient Characteristics	Whole Cohort (n = 806)	Training Dataset (n = 483) n (%)	Validation Dataset (n =121) n (%)	Test Set (n = 202) n (%)	*p*-Value
Mean age (years) (range)	59.44 ± 11.75 (23–89)	59.04 ± 11.45 (26–88)	60.03 ± 12.41 (33–89)	60.02 ± 11.66 (23–85)	0.51
Female	511 (63.40%)	306 (63.35%)	79 (65.29%)	126 (62.38%)	0.87
Non-smoker	630 (78.16%)	379 (78.47%)	92 (76.03%)	159 (78.71%)	0.88
History of alcohol consumption	87 (10.79%)	53 (10.97%)	13 (10.74%)	21 (10.40%)	0.97
Lung cancer family history	280 (34.74%)	164 (33.95%)	41 (33.88%)	75 (37.13%)	0.75
Nodule type					0.28
Solid	357 (44.29%)	217 (44.93%)	51 (42.15%)	89 (44.06%)	
Part-solid	183 (22.71%)	114 (23.60%)	25 (20.66%)	44 (21.78%)	
GGO	266 (33.00%)	152 (31.47%)	45 (37.19%)	69 (34.16%)	
Lung-RADS					0.30
2	284 (35.24%)	164 (33.96%)	47 (38.84%)	73 (36.14%)	
3	69 (8.56%)	40 (8.28%)	10 (8.27%)	19 (9.41%)	
4A	107 (13.28%)	71 (14.70%)	19 (15.70%)	17 (8.41%)	
4B, 4X	346 (42.92%)	208 (43.06%)	45 (37.19%)	93 (46.04%)	
Nodule size (cm)					0.29
<1 cm	236 (29.28%)	136 (28.16%)	38 (31.41%)	62 (30.69%)	
1–2 cm	274 (34.00%)	173 (35.82%)	44 (36.36%)	57 (28.22%)	
>2 cm	296 (36.72%)	174 (36.02%)	39 (32.23%)	83 (41.09%)	
Mean tumor size: cm (range)	2.05 ± 1.70 (0.3–10.2)	2.00 ± 1.64 (0.3–10.1)	1.92 ± 1.63 (0.5–9.6)	2.24 ± 1.87 (0.4–10.2)	0.15
Nodule location					0.16
Right upper lobe	211 (26.18%)	125 (25.88%)	35 (28.92%)	51 (25.25%)	
Right middle lobe	67 (8.31%)	49 (10.15%)	9 (7.44%)	9 (4.45%)	
Right lower lobe	171 (21.22%)	98 (20.29%)	28 (23.14%)	45 (22.28%)	
Left upper lobe	231 (28.66%)	137 (28.36%)	31 (25.62%)	63 (31.19%)	
Left lower lobe	113 (14.12%)	68 (14.08%)	18 (14.88%)	27 (13.37%)	
Others *	13 (1.61%)	6 (1.24%)	0 (0.00%)	7 (3.46%)	
Malignancy	648 (80.40%)	389 (80.54%)	96 (79.34%)	163 (80.69%)	0.95

Nodule size (%) < 1 cm, 1–2 cm, >2 cm. * Patients with a pleural lesion, a hilum lesion, or an inter-fissure lesion. Abbreviations: GGO, ground-glass opacity;

**Table 2 cancers-17-00916-t002:** Performance metrics in the test dataset according to Lung-RADS.

Lung-RADS				
	2	3	4A + 4B	4X
	(n = 73)	(n = 19)	(n = 81)	(n = 29)
AUC	0.75	0.81	0.78	0.98
Accuracy	0.84 (0.73–0.91)	0.84 (0.60–0.97)	0.82 (0.71–0.89)	0.97 (0.82–1.00)
Sensitivity	0.90 (0.79–0.96)	1.00 (0.73–1.00)	0.92 (0.82–0.97)	0.96 (0.79–1.00)
Specificity	0.57 (0.30–0.81)	0.40 (0.07–0.83)	0.44 (0.22–0.69)	1.00 (1.00–1.00)
PPV	0.90 (0.79–0.96)	0.82 (0.56–0.95)	0.85 (0.74–0.92)	1.00 (0.84–1.00)
NPV	0.57 (0.30–0.81)	1.000 (0.20–1.00)	0.62 (0.32–0.85)	0.67 (0.13–0.98)

Abbreviations: AUC, area under the receiver operating characteristic curve; NPV, negative predictive value; PPV, positive predictive value; Lung-RADS, lung imaging reporting and data system.

**Table 3 cancers-17-00916-t003:** Performance metrics in the test dataset according to tumor component.

Component			
	Solid	Part-Solid	GGO
All Nodule Sizes	(n = 89)	(n = 44)	(n = 69)
Accuracy	0.84 (0.75–0.91)	0.84 (0.70–0.93)	0.86 (0.75–0.93)
Sensitivity	0.96 (0.87–0.99)	0.89 (0.73–0.96)	0.91 (0.80–0.97)
Specificity	0.45 (0.24–0.68)	0.63 (0.26–0.90)	0.55 (0.25–0.82)
PPV	0.86 (0.76–0.92)	0.91 (0.76–0.98)	0.91 (0.80–0.97)
NPV	0.75 (0.43–0.93)	0.56 (0.23–0.85)	0.55 (0.25–0.82)

Abbreviation: GGO, ground-glass opacity; NPV, negative predictive value; PPV, positive predictive value.

**Table 4 cancers-17-00916-t004:** Performance metrics.

Epigenetic Simple Regression Model			
	Training Dataset	Validation Dataset	Test Dataset
All Nodule Sizes	(n = 483)	(n = 121)	(n = 202)
AUC	0.74	0.86	0.79
Accuracy	0.80 (0.77–0.84)	0.88 (0.81–0.94)	0.85 (0.79–0.89)
Sensitivity	0.91 (0.87–0.93)	0.95 (0.88–0.98)	0.93 (0.87–0.96)
Specificity	0.37 (0.28–0.48)	0.64 (0.43–0.81)	0.51 (0.35–0.67)
PPV ^A^	0.86 (0.82–0.89)	0.91 (0.83–0.96)	0.89 (0.83–0.93)
NPV ^A^	0.49 (0.37–0.61)	0.76 (0.53–0.91)	0.63 (0.44–0.78)
Nodules sized 5–10 mm	(n = 142)	(n = 43)	(n = 61)
AUC	0.70	0.89	0.80
Accuracy (95% CI)	0.76 (0.68–0.83)	0.88 (0.75–0.96)	0.85 (0.74–0.93)
Sensitivity (95% CI)	0.91 (0.83–0.95)	1.000 (0.86–1.000)	0.94 (0.82–0.98)
Specificity (95% CI)	0.27 (0.14–0.46)	0.62 (0.32–0.85)	0.54 (0.26–0.80)
PPV (95% CI) ^B^	0.81 (0.72–0.87)	0.86 (0.69–0.95)	0.88 (0.75–0.95)
NPV (95% CI) ^B^	0.47 (0.25–0.71)	1.000 (0.60–1.000)	0.70 (0.35–0.92)

^A^ Cancer prevalence, 80.4% in the current cohort. ^B^ Cancer prevalence, 76.0% in the current subgroup cohort. Abbreviations: AUC, area under the receiver operating characteristic curve; CI, confidence interval; NPV, negative predictive value; PPV, positive predictive value.

## Data Availability

All data generated or analyzed during this study are included in this published article and its Appendix A. Additional raw data supporting the conclusions of this article will be made available by the authors to any qualified researcher upon reasonable request. The datasets used and/or analyzed during the current study are available from the corresponding author on reasonable request.

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
