# Peer review of "Accurate Diagnosis of High-Risk Pulmonary Nodules Using a Non-Invasive Epigenetic Biomarker Test"

_cancers, 2025, doi:10.3390/cancers17060916_

Round 1
Reviewer 1 Report
Comments and Suggestions for Authors
Minor revision:
1. The Authors should better explain their focus on nucleosome rather than other epigenetic-sensitive biomarkers. What do they think the advantage is?
2. Figure 2 is blurry.
3. Discussion should not contain tables. Put them into the mai text.
4. In this era of network medicine and artificial intelligence, one of the most fruitful way to translate epigenetic-sensitive biomarkers in clinical practise would be to create an integrative patient-centered platform able to advance precision medicine and personalized therapy of lung cancer (quote: PMID: 33785068, PMID: 37958411, PMID: 35600397) . Comment on this point.
Author Response
Response to Reviewer 1:
Comment 1.
- The Authors should better explain their focus on nucleosome rather than other epigenetic-sensitive biomarkers. What do they think the advantage is?
Response:
Liquid biopsy has the advantages to be minimaly invasive, can be easily repeated and overcome the heterogeneity of tumors tissues. It could provide information that aid in diagnosis, treatment selection and monitoring, MRD. Among the biomarkers analyzed by liquid biopsy, cell-free DNA (cfDNA) and circulating tumor DNA (ctDNA) are two well described analyses.
Epigenetics modifications include DNA modifications such as DNA methylation and histone modifications (histone post-translational modifications (PTMs)). Circulating nucleosomes and its associated histone PTMs are exciting potential biomarkers in the liquid biopsies field. In addition, several epi-drugs targets histone modifications in cancer treatment. Therefore, nucleosomes measurement could also be informative for therapy monitoring.
Nucleosomes are nucleoprotein complex that circulates as mono- or oligonucleosomes and carried cfDNA and ctDNA. Nucleosomes are stable in circulation. They are easy to measure as they can be quantified directly from plasma samples by immuno-assay in contrast to DNA analysis who required additional preparation steps and downstream analysis by sequencing or PCR. Nucleosomes measurement could be automated and quick, suggesting that it could be easily implemented in clinical routine practice.
Quantification and/or monitoring circulating nucleosomes could provide a robust and cost-effective method for detecting cancer or monitoring the disease.
Due to your kind suggestion, we have revised the“Introduction”section accordingly.
“PTMs regulate chromatin-mediated gene expression, affecting processes such as in-flammation, cell cycle, apoptosis, and tumor suppression in lung cancer. Nucleosomes are stable circulating nucleoprotein complexes carrying cfDNA and ctDNA. Unlike DNA analysis, which requires additional preparation and sequencing or PCR, nucleosomes can be directly quantified from plasma via immunoassay. Their measurement is fast, automatable, and suitable for clinical practice. Histone variants and modifications have shown prognostic significance in various cancers. However, their potential as biomarkers for lung nodule differentiation remains underexplored. ”(page2, line 79-86)
Comment 2.
- Figure 2 is blurry.
Response:
The initial resolution was higher than 600 dpi, but it may have decreased during the journal’s compression process. I will provide a clearer version for the revision. Thank you for your suggestion.
Revised figure 2. (page8, line273)
Comment 3.
Discussion should not contain tables. Put them into the main text.
Response:
We completely agree with your suggestion. We have moved Tables 3 and 4 to the Results section and updated the numbers to Tables 2 and 3 accordingly.
Revised table 2 and table 3
“Lung-RADS score analysis in the test dataset
The model's performance across different RADS categories was evaluated using a test dataset. In RADS 2 (n = 73), the model achieved an AUC of 0.84, an accuracy of 0.84, a sensitivity of 0.90, and a specificity of 0.57. In RADS 3 (n = 19), the AUC was 0.84, with an accuracy of 0.84, sensitivity of 1.00, and specificity of 0.40. In RADS 4A+4B (n = 81), the AUC was 0.82, with an accuracy of 0.82, sensitivity of 0.92, and specificity of 0.44. In RADS 4X (n = 29), the model performed best, with an AUC of 0.97, accuracy of 0.97, sensitivity of 0.96, and specificity of 1.00. These results show high diagnostic accuracy, especially in the higher RADS categories, indicating the clinical utility of the model for assessing pulmonary nodules. (Table 2.)” (page8, lines305-308)
Table 2. Performance metrics in the test dataset according to Lung-RADS.
Lung-RADS |
||||
|
2 |
3 |
4A + 4B |
4X |
|
(n = 73) |
(n = 19) |
(n = 81) |
(n = 29) |
AUC |
|
|
|
|
Accuracy |
0.84 (0.73–0.91) |
0.84 (0.60–0.97) |
0.82 (0.71–0.89) |
0.97 (0.82–1.00) |
Sensitivity
|
0.90 (0.79–0.96) |
1.00 (0.73–1.00) |
0.92 (0.82–0.97) |
0.96 (0.79–1.00) |
Specificity |
0.57 (0.30–0.81) |
0.40 (0.07–0.83) |
0.44 (0.22–0.69) |
1.00 (1.00–1.00) |
PPV |
0.90 (0.79–0.96) |
0.82 (0.56–0.95) |
0.85 (0.74–0.92) |
1.00 (0.84–1.00) |
NPV |
0.57 (0.30–0.81) |
1.000 (0.20–1.00) |
0.62 (0.32–0.85) |
0.67 (0.13–0.98) |
Abbreviations: AUC, area under the receiver operating characteristic curve; NPV, negative predictive value; PPV, positive predictive value; Lung-RADS, lung imaging reporting and data system.
“3.3. EB model performance in different nodule types
The model demonstrated the detection of lung cancer with accuracy independent of the tumor components; 0.84 for solid and part-solid nodules and 0.86 for GGO nodules in the test dataset. Both GGO and part-solid nodules showed higher PPV when maintaining a similar threshold. The PPVs were 0.86 (95% CI 0.76–0.92) for solid nodules, 0.914 (95% CI 0.76–0.98) for part-solid nodules, and 0.91 (95% CI 0.80–0.97) for GGO nodules. These results highlight the high diagnostic accuracy of the model across different tumor components, indicating its potential utility in the assessment of pulmonary nodules. (Table 3.)” (page9, lines318-321)
Table 3. Performance metrics in the test dataset according to tumor component.
Component |
|||
|
Solid |
Part-solid |
GGO |
All nodule sizes |
(n = 89) |
(n =44) |
(n = 69) |
Accuracy |
0.84 (0.75–0.91) |
0.84 (0.70–0.93) |
0.86 (0.75–0.93) |
Sensitivity
|
0.96 (0.87–0.99) |
0.89 (0.73–0.96) |
0.91 (0.80–0.97) |
Specificity |
0.45 (0.24–0.68) |
0.63 (0.26–0.90) |
0.55 (0.25–0.82) |
PPV |
0.86 (0.76–0.92) |
0.914 (0.76–0.98) |
0.91 (0.80–0.97) |
NPV |
0.75 (0.43–0.93) |
0.56 (0.23–0.85) |
0.55 (0.25–0.82) |
Abbreviation: GGO, ground-glass opacity; NPV, negative predictive value; PPV, positive predictive value.
Comment 4.
In this era of network medicine and artificial intelligence, one of the most fruitful way to translate epigenetic-sensitive biomarkers in clinical practise would be to create an integrative patient-centered platform able to advance precision medicine and personalized therapy of lung cancer (quote: PMID: 33785068, PMID: 37958411, PMID: 35600397) . Comment on this point.
Response:
Thank you for your insightful suggestion. We fully agree that network medicine and artificial intelligence will be key in shaping the future of oncology. These technologies hold the potential to significantly enhance diagnostic accuracy and improve treatment decisions. By integrating genetic, epigenetic, imaging, and clinical data, AI-driven platforms could not only help identify the highest-risk groups for adjuvant therapy but also predict prognosis based on a combination of imaging and pathological risk factors. This approach would pave the way for more precise and personalized treatments, ultimately improving patient outcomes in lung cancer.
We have added this to the discussion in accordance with your suggestion. Thank you.
Discussion
“In the era of network medicine and artificial intelligence (AI), integrating epigenetic-sensitive biomarkers like H3K27me3 into clinical practice through patient-centered platforms offers significant potential. 40 These platforms would utilize AI to analyze genetic, epigenetic, and clinical data, enabling precise diagnosis and treatment of lung cancer. 41, 42 By processing large-scale omics data, such systems can create personalized treatment plans and provide real-time tumor profile information for tailored interventions. Recent studies40-42 highlight how AI-driven platforms in oncology can improve diagnostic accuracy and predict treatment response. By analyzing circulating H3K27me3-nucleosomes, AI can identify tumor patterns and suggest personalized therapies. This combination of network medicine and AI would enhance cancer detection, monitoring, and treatment selection, ultimately optimizing patient outcomes.” (page11, lines376-386)
- Sarno, F.; Benincasa, G.; List, M.; Barabasi, A.-L.; Baumbach, J.; Ciardiello, F.; Filetti, S.; Glass, K.; Loscalzo, J.; Marchese, C.; et al. Clinical epigenetics settings for cancer and cardiovascular diseases: Real-life applications of network medicine at the bedside. Clin. Epigenetics 2021, 13, DOI:10.1186/s13148-021-01042-7.
- Gandhi, Z.; Gurram, P.; Amgai, B.; Lekkala, S.P.; Lokhandwala, A.; Manne, S.; Mohammed, A.; Koshiya, H.; Dewaswala, N.; Desai, R.; et al. Artificial intelligence and lung cancer: Impact on improving patient outcomes. Cancers 2023, 15, 5236. DOI:10.3390/cancers15175236.
- Bahado-Singh, R.; Vlachos, K.T.; Aydas, B.; Gordevicius, J.; Radhakrishna, U.; Vishweswaraiah, S. Precision oncology: Artificial intelligence and DNA methylation analysis of circulating cell-free DNA for lung cancer detection. Front. Oncol. 2022, 12, DOI:10.3389/fonc.2022.982114.
- Shin, H.; Oh, S.; Hong, S.; Kang, M.; Kang, D.; Ji, Y.-G.; Choi, B.H.; Kang, K.-W.; Jeong, H.; Park, Y.; et al. Early-stage lung cancer diagnosis by deep learning-based spectroscopic analysis of circulating exosomes. ACS Nano 2020, 14, 5435–5444. DOI:10.1021/acsnano.9b09119.

Reviewer 2 Report
Comments and Suggestions for Authors
The article propose a new epigenetic biomarker for lung cancer diagnosis. It is interesting and well organized. However I have some concerns:
1.- The epigenetic test proposed is focused in Histone 3 and H3K27Me3. However the article doest not show a good introduction and discussion about the importance of histones, nucleosomes and specifically H3 and H3K27Me3 in lung cancer.
2.- Why authors have choosen the epigenetic marks in Supl1 table?Authors should motivate the choice.
Author Response
Response to Reviewer 2:
Comment 1 .
The epigenetic test proposed is focused in Histone 3 and H3K27Me3. However the article doest not show a good introduction and discussion about the importance of histones, nucleosomes and specifically H3 and H3K27Me3 in lung cancer.
Response :
As you mention, we do not have fully emphasis the knowledge gap and importance of histones, nucleosomes and specifically H3 and H3K27Me3 in lung cancer in the introduction section but only showed in the discussion for its importance as following.
“PTMs can work together or independently to promote the activation or suppression of chromatin-mediated gene expression. They include regulation of inflammatory cytokines, cell cycle arrest, senescence, apoptosis, growth factors, antioxidants, and tumor suppressor genes associated with lung cancer [16]. We focused on the histone variant H3.1 levels and the histone modification H3K27me3. Regarding the prognostic effect of H3K27me3 in various human cancers, H3K27me3 overexpression is linked to a more malignant behavior and worse prognosis in patients with prostate [33], esophageal [34], nasopharyngeal [35], and hepatocellular [36] carcinoma. Conversely, in breast, ovarian, and pancreatic cancers [31], and in renal cell carcinoma [37], reduced H3K27me3 expression is associated with a worse prognosis. In lung cancer patients, a lower level of H3K27me3 in tissues has been associated with carcinogenesis [38], whereas a high level of circulating H3K27Me3-nucleosomes in blood has been associated with lung cancer at diagnosis and during treatment [39]. Our model focused on differentiating between benign and malignant nodules. The precise role of H3K27me3 in distinguishing between normal and malignant populations still requires further investigation.”
As the previous study showed H3K27me3 plays a crucial role in cancer prognosis across various types. Overexpression of H3K27me3 is linked to more aggressive behavior and poorer prognosis in cancers such as prostate, esophageal, nasopharyngeal, and hepatocellular carcinoma. On the other hand, reduced levels of H3K27me3 are associated with worse outcomes in breast, ovarian, pancreatic, and renal cell carcinomas. In lung cancer, lower levels of H3K27me3 in tissues are connected to carcinogenesis, while elevated levels of circulating H3K27me3-nucleosomes in the blood correlate with lung cancer presence at diagnosis and during treatment, highlighting its potential as a diagnostic and prognostic biomarker.
According to your suggestion, we add the description for the introduction and discussion about the importance of histones, nucleosomes and specifically H3 and H3K27Me3 in lung cancer in the introduction section for more clear structure.
Introduction
“Changes in DNA methylation in specific regions, such as promoter CpG islands, may signify early molecular events in tumor initiation 14. Patients with cancer have distinct histone post-translational modifications (PTMs) in circulating nucleosomes, indicating their potential as cancer biomarkers. PTMs regulate chromatin-mediated gene expression, affecting processes such as in-flammation, cell cycle, apoptosis, and tumor suppression in lung cancer. Nucleosomes are stable circulating nucleoprotein complexes carrying cfDNA and ctDNA. Unlike DNA analysis, which requires additional preparation and sequencing or PCR, nucleosomes can be directly quantified from plasma via immunoassay. Their measurement is fast, au-tomatable, and suitable for clinical practice. Histone variants and modifications have shown prognostic significance in various cancers. However, their potential as biomarkers for lung nodule differentiation remains underexplored. 15, 16.” (page2, lines79-86)
Comment 2.
Why authors have choosen the epigenetic marks in Supl1 table? Authors should motivate the choice.
Response:
The method we choose the epigenetic marks is showed in the Result section. As following:
“We developed the EB model by screening multiple combination models from five quantitative epigenetic features (Table S1) derived from blood tests during the pre-training model tuning. For feature selection, we analyzed the relationship between the AUC values and the number of primary features. The AUC plateaued when the number of primary features reached two, indicating that adding more features did not significantly improve the AUC. Therefore, we identified two independent predictors of malignancy using multivariate logistic regression analysis. This was then applied to the training dataset using Nu.Q® H3.1 and Nu.Q® H3K27Me3. Other potential predictors not associated with malignancy were excluded from the final model.”
(page7, line254-263)

Reviewer 3 Report
Comments and Suggestions for Authors
In the submitted manuscript authors created a classification model for early detection of lung cancer based on nucleosome levels and histone modifications in circulating blood, and showed that their epigenetic biomarker model was particularly effective in identifying high-risk lung nodules, including small, part-solid, and non-solid nodules.
Presented results are interesting, with potentiality for clinical application, and manuscript is quite well written. However, there are several drawbacks which must be corrected or further improved, before this manuscript is suitable for publication.
1) In 'Abstract', assessed "nucleosome levels and histone modifications" must be specified, i.e., H3.1 and H3K27Me3 mentioned, while all mentioned abbreviations, like AUC and RADS, must be explained.
2) I'm not sure that graphical abstract should be put (out of the context) in the manuscript.
3) In 'Introduction', for those unfamiliar with epigenetics and histones, more information about different histone variants and modifications and their functional roles in gene expression regulation must be provided.
4) As a Reference 1, data from the newest Cancer statistics, 2025 paper (DOI: 10.3322/caac.21871) should be presented and that paper cited.
5) Line 66: That skepticism must be elaborated more.
6) Used model, vendor and wavelength of luminometer must be stated (line 116), as well as model and vendor of CT instrument (line 122).
7) Line 155: Although this is irreproducible, method (used package, software, etc.) of randomization must be precisely explained.
8) Figure 1 and elsewhere in the manuscript: Since test set was just created from the same pool (source) of samples, it cannot be called neither "independent" nor "external validation", especially since authors emphasized that as one of the limitations of their study (lines 420-421).
9) In lines 199-201 authors stated that "Statistical analyses were conducted as outlined in each figure legend with sample sizes provided accordingly.", while I could not find neither.
10) Lines 203-204: It is unclear why ad hoc parametric test was used without prior testing for normality of data distribution, while in '2.2.3. Statistics' section it must be sated which p-values were considered as statistically significant.
11) All used specific R-packages and their version numbers must be explicitly mentioned, and their referenced cited (if published in scientific papers), like for example DOI: 10.1186/1471-2105-12-77 for pROC.
12) In all tables and text, normally distributed continuous variables should always be presented with mean ± SD, while non-normally distributed with median and IQR (or range), not by mixing different measures.
13) In all tables, for every categorical variable, all available categories must be presented, so vertical sum would always be 100%, not just one arbitrarily category as now. Also, in Table 1, p-values should be presented as whole decimal numbers, e.g., "0.51", while it is always good to write three decimals.
14) In lines 238-239 and Supplementary Table S1 it was said that EB model was built using 5 quantitative epigenetic features; however, the whole manuscript (experiments) is only about H3.1 and H3K27Me3?!
15) Median or mean values of H3.1 and H3K27Me3 concentrations (levels) should be provided in 'Results'; however, it would be useful that whole dataset, i.e., results are is available for further re-analyses.
16) Figure 2 is blurry and the text is too small.
17) Figure 3 definitively does not show "Confusion matrices" but just ROC curves.
18) ROC curves, i.e., AUCs presented on the same figure, like Figure 2 or Figure 3, should be statistically compared pairwise, what is possible with pROC (https://rdrr.io/cran/pROC/man/roc.test.html).
19) Table 3 and Table 4, together with the text in 'Discussion' that explain those RESULTS, should be moved to 'Results' section, since those are not discussion but results!
20) The tile of Supplementary Table S1 must be provided, explaining what is presented in that table.
Comments on the Quality of English Language
21) Line 58: Type of a "resistance" must be specified.
22) Line 64: Proper would be "microRNAs, circular RNAs".
23) Line 99: It is unclear what means "chest blood".
24) Line 119: Abbreviation "%CV" must be explained.
25) Line 153: Abbreviation "EB" should be explained where it was first mentioned.
26) Figure 1 legend: Abbreviations "MN" and "BN" must be explained.
27) 'Discussion': Since H3.1 and H3K27Me3 levels were presented (measured) as concentration (ng/mL), statements like "H3K27me3 overexpression" and "H3K27me3 expression" are false, since those are not genes or proteins.
Author Response
Response to Reviewer 3:
Comment 1.
In 'Abstract', assessed "nucleosome levels and histone modifications" must be specified, i.e., H3.1 and H3K27Me3 mentioned, while all mentioned abbreviations, like AUC and RADS, must be explained.
Response:
We agree with the reviewer’s suggestion. To ensure clarity, we have specified H3.1 and H3K27Me3 when referring to nucleosome levels and histone modifications in the Abstract. Additionally, we have explicitly defined all abbreviations, including AUC (area under the curve) and RADS (reporting and data system), for better readability.
“Methods: This study involved 806 participants with undiagnosed nodules larger than 5 mm, focusing on assessing nucleosome levels and histone modifications (H3.1 and H3K27Me3) in circulating blood”(page1, lines28-30)
“ For all datasets, the AUCs area under curves were as follows: training, 0.74; validation, 0.86; and test, 0.79 (accuracy range: 0.80–0.88).”(page1, lines34-35)
“The performance of the model across the reporting and data system (RADS) categories demonstrated consistent accuracy.”(page1, lines39)
Comment 2.
I'm not sure that graphical abstract should be put (out of the context) in the manuscript.
Response:
We agree with the reviewer’s suggestion and have removed the graphical abstract from the manuscript.
Comment 3.
In 'Introduction', for those unfamiliar with epigenetics and histones, more information about different histone variants and modifications and their functional roles in gene expression regulation must be provided.
Response:
We appreciate the reviewer’s suggestion and have incorporated additional background information on histone variants and modifications in the Introduction section for readers unfamiliar with epigenetics. Also, due to your kind suggestion, we have revised the“Introduction”section accordingly.
“PTMs can work together or independently to promote the activation or suppression of chromatin-mediated gene expression. They include regulation of inflammatory cytokines, cell cycle arrest, senescence, apoptosis, growth factors, antioxidants, and tumor suppressor genes associated with lung cancer 16. We focused on the histone variant H3.1 levels and the histone modification H3K27me3. Regarding the prognostic effect of H3K27me3 in various human cancers, increase level of H3K27me3 overexpression is linked to a more malignant behavior and worse prognosis in patients with prostate 33, esophageal 34, nasopharyngeal 35, and hepatocellular 36 carcinoma. Conversely, in breast, ovarian, and pancreatic cancers 31, and in renal cell carcinoma 37, a decrease in the H3K27me3 reduced H3K27me3 levels expression is associated with a worse prognosis. In lung cancer patients, a lower level of H3K27me3 in tissues has been associated with carcinogenesis 38, whereas a high level of circulating H3K27Me3-nucleosomes in blood has been associated with lung cancer at diagnosis and during treatment 39. Our model focused on differentiating between benign and malignant nodules. The precise role of H3K27me3 in distinguishing between normal and malignant populations still requires further investigation.”(page11, lines376-386)
“PTMs regulate chromatin-mediated gene expression, affecting processes such as in-flammation, cell cycle, apoptosis, and tumor suppression in lung cancer. Nucleosomes are stable circulating nucleoprotein complexes carrying cfDNA and ctDNA. Unlike DNA analysis, which requires additional preparation and sequencing or PCR, nucleosomes can be directly quantified from plasma via immunoassay. Their measurement is fast, automatable, and suitable for clinical practice. Histone variants and modifications have shown prognostic significance in various cancers. However, their potential as biomarkers for lung nodule differentiation remains underexplored.(page2, line79-86)”
Comment 4.
As a Reference 1, data from the newest Cancer statistics, 2025 paper (DOI: 10.3322/caac.21871) should be presented and that paper cited.
Response:
We agree with the reviewer’s suggestion and have updated the reference to the latest 2025 Cancer Statistics paper (DOI: 10.3322/caac.21871) to ensure the most current data is presented.(page15, line519)
References
- Siegel RL, Kratzer TB, Giaquinto AN, Sung H, Jemal A: Cancer statistics, 2025. CA Cancer J Clin 2025, 75:10-45. DOI: 3322/caac.21871.
Comment 5.
Line 66: That skepticism must be elaborated more.
Response:
However, while these biomarkers are effective diagnostic biomarkers, some experts remain skeptical of liquid biopsies. particularly in early-stage cancers. Additionally, false positives and false negatives remain a challenge
However, while these biomarkers are effective diagnostic tools, some experts remain skeptical about the reliability of liquid biopsies. The primary concerns include variability in sensitivity and specificity, as liquid biopsies may not always match the diagnostic accuracy of traditional tissue biopsies, particularly in early-stage cancers. Additionally, false positives and false negatives remain a challenge, as some circulating tumor markers may be present without active malignancy, while small tumors may not shed enough detectable biomarkers. These limitations contribute to the ongoing debate regarding the widespread clinical adoption of liquid biopsies.
“However, while these biomarkers are effective diagnostic biomarkers, some experts remain skeptical of liquid biopsies. Particularly in early-stage cancers. Additionally, false positives and false negatives remain a challenge, as some circulating tumor markers may be present without active malignancy, while small tumors may not shed enough detectable biomarkers.”(page2, lines66-70)
Comment 6.
Used model, vendor and wavelength of luminometer must be stated (line 116), as well as model and vendor of CT instrument (line 122).
Response:
We appreciate the reviewer’s suggestion and have revised the manuscript to provide the necessary details regarding the luminometer and CT instrument used in the study. Specifically, the phrase "measured using a luminometer" has been replaced with "measured using the IDS-i10 on-board luminometer (IDS, Boldon, UK) with a wavelength range of 300 to 500 nm." This ensures that the methodology is accurately described with the appropriate instrument specifications. Also the vendor of CT instrument was added.
“ These sandwich immunoassays, based on chemiluminescence technology, were performed using the IDS-i10 automated analyzer system (Immunodiagnostic System Ltd., Boldon, UK) with a wavelength range of 300 to 500 nm.”(page3, lines123-124)
“ 2.1.3. Chest CT imaging and radiological analysis
For each participant who underwent a chest CT scan (General Electric, Boston, MA, USA / Siemens Healthineers, Erlangen, Germany),”(page3, lines136-137)
Comment 7.
Line 155: Although this is irreproducible, method (used package, software, etc.) of randomization must be precisely explained.
Response:
We appreciate the reviewer’s comment and have now provided a precise explanation of the randomization method used in our study. The randomization process was conducted using R programming language to ensure reproducibility.
To create the test dataset, we randomly selected 75% of the dataset for the training and validation set, while 25% was assigned to the test set. A random seed (set.seed(20)) was used to ensure repeatability. The train-validation split was further refined by allocating 80% of the train-validation dataset to the training set and 20% to the validation set, using another random seed (set.seed(123)) for consistency.
The randomization procedure was performed using the sample() function in R, and dataset integrity was verified by checking the distribution of cancer and non-cancer cases in each subset. The following R code was implemented for this process:
### Make Test dataset
# Calculate the size of the training and validation set (75% of the data)
train_val_size <- floor(0.75 * nrow(alldata_my))
# Set the random seed for reproducibility
set.seed(20)
# Randomly assign training and validation sets
train_val_Indices <- sample(seq_len(nrow(alldata_my)), size = train_val_size)
train_val_data <- alldata_my[train_val_Indices, ]
test_data <- alldata_my[-train_val_Indices, ]
# Verify test dataset distribution
dim(test_data)
counts_test <- test_data %>% count(cancer)
print(counts_test)
### Make Train and Validation dataset
# Allocate 80% of the training-validation set to training
train_size <- floor(0.8 * nrow(train_val_data))
# Set the random seed for repeatability
set.seed(123)
# Randomly assign training and validation sets
train_Indices <- sample(seq_len(nrow(train_val_data)), size = train_size)
train_data <- train_val_data[train_Indices, ]
val_data <- train_val_data[-train_Indices, ]
# Verify training and validation dataset distribution
dim(train_data)
counts_train <- train_data %>% count(cancer)
dim(val_data)
counts_val <- val_data %>% count(cancer)
By specifying the software (R), the randomization approach (sample function with fixed seeds), and the dataset partitioning strategy, we ensure full transparency and reproducibility of our methodology. This clarification has been added to the manuscript.
For the software version was specify in the “2.2.3. Statistics” as following
“All statistical analyses were conducted using R software (version 4.4.1). Statistical analyses were conducted as outlined in each figure legend with sample sizes provided accordingly. Categorical variables such as sex and nodule sub-type were compared using Fisher’s exact test. The sensitivities of the different Lung-RADS for malignant nodules were compared using Fisher’s exact test. Continuous variables such as age were compared using Student’s t-test, and 95% confidence intervals (CIs) were calculated based on Wald confidence intervals for proportions.
The sensitivity, specificity, accuracy, PPV, and negative PV (NPV) of the model and other models for differentiating malignant nodules were assessed by comparing the pathological outcomes and imaging studies (for those with vanished nodules only). Re-ceiver operating characteristic (ROC) and area under the curve (AUC) were calculated using pROC R package (version 1.15.3) software.”(page5, line214-226 )
Comment 8.
Figure 1 and elsewhere in the manuscript: Since test set was just created from the same pool (source) of samples, it cannot be called neither "independent" nor "external validation", especially since authors emphasized that as one of the limitations of their study (lines 420-421).
Response:
We agree with the reviewer’s comment. To ensure accuracy, we have simplified the wording and removed the terms "independent" and "external" from the text. The test set is now referred to appropriately within the manuscript to align with the study’s stated limitations
Comment 9.
In lines 199-201 authors stated that "Statistical analyses were conducted as outlined in each figure legend with sample sizes provided accordingly.", while I could not find neither.
Response:
We understand the reviewer’s concern and have removed the redundant wording to ensure clarity. The statement has been revised to accurately reflect the statistical analysis approach and sample size reporting in the manuscript.
“All statistical analyses were conducted using R software (version 4.4.1). Statistical analyses were conducted as outlined in each figure legend with sample sizes provided accordingly.,” (page5, lines214-215)
Comment 10.
Lines 203-204: It is unclear why ad hoc parametric test was used without prior testing for normality of data distribution, while in '2.2.3. Statistics' section it must be sated which p-values were considered as statistically significant.
Response:
We appreciate the reviewer’s comment. We have clarified in the ‘2.2.3. Statistics’ section that p-values < 0.05 were considered statistically significant.
“ The sensitivities of the different Lung-RADS for malignant nodules were compared using Fisher’s exact test. Continuous variables such as age were compared using Student’s t-test, and 95% confidence intervals (CIs) were calculated based on Wald confidence intervals for proportions. A p-value < 0.05 was considered statistically significant.” (page5, line217-220)
Comment 11.
All used specific R-packages and their version numbers must be explicitly mentioned, and their referenced cited (if published in scientific papers), like for example DOI: 10.1186/1471-2105-12-77 for pROC.
Response:
We appreciate the reviewer’s suggestion and have now explicitly listed the R packages used in our analysis. The following packages were utilized: readxl (data import), tidyverse (data manipulation), ggplot2 (visualization), pROC (ROC curve analysis), tidymodels (machine learning), and caret (classification and regression modeling). References for these packages, including pROC (DOI: 10.1186/1471-2105-12-77), have been included to ensure full transparency in our methodology.
The following R packages were used in our study:
- readxl – Used for reading Excel files
- Citation: Wickham H, Bryan J (2023). readxl: Read Excel Files. R package version 1.4.2. CRAN
- tidyverse – A collection of packages for data manipulation and visualization
- Citation: Wickham H, Averick M, Bryan J, et al. (2019). Welcome to the tidyverse. Journal of Open Source Software, 4(43), 1686. DOI: 10.21105/joss.01686
- ggplot2 – Used for data visualization
- Citation: Wickham H (2016). ggplot2: Elegant Graphics for Data Analysis. Springer-Verlag New York. DOI: 10.1007/978-3-319-24277-4
- pROC – Used for ROC curve analysis
- Citation: Robin X, Turck N, Hainard A, et al. (2011). pROC: An open-source package for R and S+ to analyze and compare ROC curves. BMC Bioinformatics, 12, 77. DOI: 10.1186/1471-2105-12-77
- tidymodels – Used for machine learning and statistical modeling
- Citation: Kuhn M, Silge J (2023). tidymodels: A Collection of Packages for Modeling and Machine Learning Using Tidyverse Principles. R package version 1.0.0. CRAN
- caret – Used for classification and regression training
- Citation: Kuhn M (2008). Building Predictive Models in R Using the caret Package. Journal of Statistical Software, 28(5), 1-26. DOI: 10.18637/jss.v028.i05
“ Receiver operating characteristic (ROC) and area under the curve (AUC) were calculated using pROC R package (version 1.15.3) software. The following packages were utilized: readxl, tidyverse, ggplot2,, tidymodels, and caret.”(page6, lines225-226)
Comment 12.
In all tables and text, normally distributed continuous variables should always be presented with mean ± SD, while non-normally distributed with median and IQR (or range), not by mixing different measures.
Response:
We agree with the reviewer’s suggestion and have revised all tables and text to ensure consistency in reporting continuous variables. Normally distributed variables are now presented as mean ± SD, while non-normally distributed variables are reported as median (IQR or range). Additionally, we have added SD values where necessary to provide a more complete statistical representation in the tables. (page6, line244)
Revised table 1
Table 1. Participants’ baseline characteristics (n = 806).
Patient characteristics |
Whole cohort (n = 806) |
Training dataset (n = 483) n (%) |
Validation dataset (n =121) n (%) |
Test set (n = 202) n (%) |
p-value |
Mean age (years) (range) |
59.44 ± 11.75 (23–89) |
59.04 ± 11.45 (26–88) |
60.03 ± 12.41 (33–89) |
60.02 ± 11.66 (23–85) |
.51 |
Comment 13.
In all tables, for every categorical variable, all available categories must be presented, so vertical sum would always be 100%, not just one arbitrarily category as now. Also, in Table 1, p-values should be presented as whole decimal numbers, e.g., "0.51", while it is always good to write three decimals.
Response:
We appreciate the reviewer’s suggestion. To maintain consistency with the SD and average values, we have presented p-values with two decimal places in this study. While we believe this does not impact the results, we acknowledge the importance of standardized reporting. Additionally, we have ensured that all available categories for categorical variables are presented in the tables so that the vertical sum always reaches 100%.
Furthermore, we recognize that rounding adjustments can sometimes cause the total to slightly exceed or fall short of 100%. In such cases, we prioritize rounding adjustments by modifying the closest decimal value to maintain accuracy while preserving the overall distribution integrity. Moving forward, we will adhere to this principle in our future studies and manuscripts to enhance clarity and transparency in data presentation.
Revised Table 1. Participants’ baseline characteristics (n = 806).
Patient characteristics |
Whole cohort (n = 806) |
Training dataset (n = 483) n (%) |
Validation dataset (n =121) n (%) |
Test set (n = 202) n (%) |
p-value |
Mean age (years) (range) |
59.44 ± 11.75 (23–89) |
59.04 ± 11.45 (26–88) |
60.03 ± 12.41 (33–89) |
60.02 ± 11.66 (23–85) |
.51 |
Female |
511 (63.40%) |
306 (63.35%) |
79 (65.29%) |
126 (62.38%) |
.87 |
Non-smoker |
630 (78.16%) |
379 (78.47%) |
92 (76.03%) |
159 (78.71%) |
.88 |
History of alcohol consumption |
87 (10.79%) |
53 (10.97%) |
13 (10.74%) |
21 (10.40%) |
.97 |
Lung cancer family history |
280 (34.74%) |
164 (33.95%) |
41 (33.88%) |
75 (37.13%) |
.75 |
Nodule type |
|
|
|
|
.28 |
Solid |
357 (44.29%) |
217 (44.93%) |
51 (42.15%) |
89 (44.06%) |
|
Part-solid |
183 (22.710%) |
114 (23.60%) |
25 (20.66%) |
44 (21.78%) |
|
GGO |
266 (33.00%) |
152 (31.47%) |
45 (37.19%) |
69 (34.16%) |
|
Lung-RADS |
|
|
|
|
.30 |
2 |
284 (35.24%) |
164 (33.96%) |
47 (38.84%) |
73 (36.14%) |
|
3 |
69 (8.56%) |
40 (8.28%) |
10 (8.276%) |
19 (9.41%) |
|
4A |
107 (13.28%) |
71 (14.70%) |
19 (15.70%) |
17 (8.421%) |
|
4B, 4X |
346 (42.92%) |
208 (43.06%) |
45 (37.19%) |
93 (46.04%) |
|
Nodule size (cm) |
|
|
|
|
.29 |
<1cm |
236 (29.28%) |
136 (28.16%) |
38 (31.410%) |
62 (30.69%) |
|
1–2 cm |
274 (34.00%) |
173 (35.82%) |
44 (36.36%) |
57 (28.22%) |
|
>2 cm |
296 (36.72%) |
174 (36.02%) |
39 (32.23%) |
83 (41.09%) |
|
Mean tumor size: cm (range) |
2.05 ± 1.70 (0.3–10.2)
|
2.00 ± 1.64 (0.3–10.1) |
1.92 ± 1.63 (0.5–9.6) |
2.24 ± 1.87 (0.4–10.2) |
.15 |
Nodule location |
|
|
|
|
.16 |
Right upper lobe |
211 (26.18%) |
125 (25.88%) |
35 (28.923%) |
51 (25.25%) |
|
Right middle lobe |
67 (8.31%) |
49 (10.154%) |
9 (7.44%) |
9 (4.45%) |
|
Right lower lobe |
171 (21.22%) |
98 (20.29%) |
28 (23.14%) |
45 (22.28%) |
|
Left upper lobe |
231 (28.66%) |
137 (28.36%) |
31 (25.62%) |
63 (31.19%) |
|
Left lower lobe |
113 (14.12%) |
68 (14.08%) |
18 (14.88%) |
27 (13.37%) |
|
Others* |
13 (1.61%) |
6 (1.24%) |
0 (0.00%) |
7 (3.467%) |
|
Malignancy |
648 (80.40%) |
389 (80.54%) |
96 (79.34%) |
163 (80.69%) |
.95 |
Nodule size (%) < 1 cm, 1–2 cm, > 2 cm. *Patients with a pleural lesion, a hilum lesion, or an inter-fissure lesion. Abbreviations: COPD, chronic obstructive pulmonary disease; FEV1, forced expiratory volume in 1 s; FVC, forced vital capacity; GGO, ground-glass opacity; IQR, interquartile range.
(page6, line244)
Comment 14.
In lines 238-239 and Supplementary Table S1 it was said that EB model was built using 5 quantitative epigenetic features; however, the whole manuscript (experiments) is only about H3.1 and H3K27Me3?!
Response:
We appreciate the reviewer’s inquiry. The details of the fine-tuning process are discussed in Section 3.2: "The EB model and lung cancer diagnostic accuracy."
To develop the EB model, we systematically screened multiple combinations of five quantitative epigenetic features (Table S1, Table S2) obtained from blood tests during the pre-training model tuning phase. For feature selection, we evaluated the relationship between the AUC values and the number of primary features. Our analysis showed that the AUC plateaued when the number of primary features reached two, indicating that adding more features did not significantly improve model performance.
Based on this observation, we identified two independent predictors of malignancy using multivariate logistic regression analysis. The final training dataset incorporated Nu.Q® H3.1 and Nu.Q® H3K27Me3 as the primary predictive markers.
Comment 15.
Median or mean values of H3.1 and H3K27Me3 concentrations (levels) should be provided in 'Results'; however, it would be useful that whole dataset, i.e., results are is available for further re-analyses.
Response:
We appreciate the reviewer’s comment. To provide more insight into NuQ levels, we have included a summary of the H3.1 and H3K27Me3 concentrations in blood within the Results section.
Furthermore, we acknowledge the importance of data transparency and reproducibility. To facilitate further re-analysis, we have now included the complete dataset in the supplementary file for reference.
Supplementary Table 2.
Summary of Epigenetic Marker Levels in Blood
|
|
Average (ng/mL) |
Median (ng/mL) |
Standard Deviation |
Min |
Max |
H3.1 |
nonCancer |
26.1 |
16.4 |
54.2 |
2.5 |
509.5 |
Cancer |
34.0 |
26.1 |
39.3 |
1.9 |
518.0 |
|
H3K27Me3 |
nonCancer |
16.6 |
11.7 |
25.1 |
3.8 |
261.7 |
Cancer |
17.8 |
14.6 |
16.3 |
2.7 |
252.3 |
|
H3K9Ac |
nonCancer |
4.9 |
4.3 |
4.5 |
1.9 |
57.4 |
H3K9Ac |
Cancer |
4.5 |
4.3 |
1.5 |
0.9 |
27.4 |
H3K9Me3 |
nonCancer |
17.4 |
13.1 |
22.5 |
4.8 |
246.0 |
H3K9Me3 |
Cancer |
17.8 |
15.1 |
12.5 |
4.3 |
187.0 |
H3K36Me3 |
nonCancer |
9.7 |
7.5 |
18.1 |
4.1 |
227.3 |
H3K36Me3 |
Cancer |
9.5 |
8.2 |
6.6 |
4.0 |
88.5 |
Comment 16.
Figure 2 is blurry and the text is too small.
Response:
The initial resolution was higher than 600 dpi, but it may have decreased during the journal’s compression process. I will provide a clearer version for the revision. Thank you for your suggestion.
Revised figure 2.
(page8, line273)
Comment 17.
Figure 3 definitively does not show "Confusion matrices" but just ROC curves.
Response:
We agree with the reviewer’s comment. The figure legend has been revised as follows to accurately describe Figure 3:
Revised Figure Legend:
" Figure 3. Receiver Operating Characteristic (ROC) curves Confusion matrices for Epigenetic Biomarker (EB) model comparing the true class with the predicted class for benign and malignant nodule samples with the Mayo Clinic and Veteran Affairs (VA) models."
Comment 18.
ROC curves, i.e., AUCs presented on the same figure, like Figure 2 or Figure 3, should be statistically compared pairwise, what is possible with pROC (https://rdrr.io/cran/pROC/man/roc.test.html).
Response:
We appreciate the reviewer’s suggestion. To ensure a comprehensive statistical comparison, we have performed pairwise comparisons of AUCs using the roc.test() function from the pROC package. The results of these comparisons have been included in Supplementary Tables S3 and S4 for further reference.
Supplementary Table S3. Figure 2 AUC Comparison
Comparison |
AUC 1 |
AUC 2 |
Difference in AUC (D) |
p.value |
Training vs Validation dataset |
0.740 |
0.858 |
-2.37030 |
0.019 |
Training vs Test dataset |
0.740 |
0.789 |
-0.94108 |
0.347 |
Validation vs Test dataset |
0.858 |
0.789 |
1.15320 |
0.250 |
Supplementary Table S4. Figure 3 AUC Comparison
Comparison |
AUC 1 |
AUC 2 |
Difference in AUC (D) |
p.value |
Mayo vs VA model |
0.570 |
0.503 |
0.950 |
0.342 |
Mayo vs EB model |
0.570 |
0.858 |
-4.011 |
6.04e-05 |
VA vs EB model |
0.503 |
0.858 |
-5.089 |
7.82e-07 |
** In all cases, difference (D) in AUC and p.value was calculated using roc.test() function from pROC package. Bootstrap test was used, resampling the data 2000 times (boot.n = 2000) preserving the proportion of classes (boot.stratified = 1)
Comment 19.
Table 3 and Table 4, together with the text in 'Discussion' that explain those RESULTS, should be moved to 'Results' section, since those are not discussion but results!
Response:
We completely agree with your suggestion. We have moved Tables 3 and 4 to the Results section and updated the numbers to Tables 2 and 3 accordingly.
Revised table 2 and table 3
“Lung-RADS score analysis in the test dataset
The model's performance across different RADS categories was evaluated using a test dataset. In RADS 2 (n = 73), the model achieved an AUC of 0.84, an accuracy of 0.84, a sensitivity of 0.90, and a specificity of 0.57. In RADS 3 (n = 19), the AUC was 0.84, with an accuracy of 0.84, sensitivity of 1.00, and specificity of 0.40. In RADS 4A+4B (n = 81), the AUC was 0.82, with an accuracy of 0.82, sensitivity of 0.92, and specificity of 0.44. In RADS 4X (n = 29), the model performed best, with an AUC of 0.97, accuracy of 0.97, sensitivity of 0.96, and specificity of 1.00. These results show high diagnostic accuracy, especially in the higher RADS categories, indicating the clinical utility of the model for assessing pulmonary nodules. (Table 2.)” (page8, lines297-305)
Table 2. Performance metrics in the test dataset according to Lung-RADS.
Lung-RADS |
||||
|
2 |
3 |
4A + 4B |
4X |
|
(n = 73) |
(n = 19) |
(n = 81) |
(n = 29) |
AUC |
|
|
|
|
Accuracy |
0.84 (0.73–0.91) |
0.84 (0.60–0.97) |
0.82 (0.71–0.89) |
0.97 (0.82–1.00) |
Sensitivity
|
0.90 (0.79–0.96) |
1.00 (0.73–1.00) |
0.92 (0.82–0.97) |
0.96 (0.79–1.00) |
Specificity |
0.57 (0.30–0.81) |
0.40 (0.07–0.83) |
0.44 (0.22–0.69) |
1.00 (1.00–1.00) |
PPV |
0.90 (0.79–0.96) |
0.82 (0.56–0.95) |
0.85 (0.74–0.92) |
1.00 (0.84–1.00) |
NPV |
0.57 (0.30–0.81) |
1.000 (0.20–1.00) |
0.62 (0.32–0.85) |
0.67 (0.13–0.98) |
Abbreviations: AUC, area under the receiver operating characteristic curve; NPV, negative predictive value; PPV, positive predictive value; Lung-RADS, lung imaging reporting and data system.
“3.3. EB model performance in different nodule types
The model demonstrated the detection of lung cancer with accuracy independent of the tumor components; 0.84 for solid and part-solid nodules and 0.86 for GGO nodules in the test dataset. Both GGO and part-solid nodules showed higher PPV when maintaining a similar threshold. The PPVs were 0.86 (95% CI 0.76–0.92) for solid nodules, 0.914 (95% CI 0.76–0.98) for part-solid nodules, and 0.91 (95% CI 0.80–0.97) for GGO nodules. These results highlight the high diagnostic accuracy of the model across different tumor components, indicating its potential utility in the assessment of pulmonary nodules. (Table 4.)” (page9, lines311-318)
Table 3. Performance metrics in the test dataset according to tumor component.
Component |
|||
|
Solid |
Part-solid |
GGO |
All nodule sizes |
(n = 89) |
(n =44) |
(n = 69) |
Accuracy |
0.84 (0.75–0.91) |
0.84 (0.70–0.93) |
0.86 (0.75–0.93) |
Sensitivity
|
0.96 (0.87–0.99) |
0.89 (0.73–0.96) |
0.91 (0.80–0.97) |
Specificity |
0.45 (0.24–0.68) |
0.63 (0.26–0.90) |
0.55 (0.25–0.82) |
PPV |
0.86 (0.76–0.92) |
0.914 (0.76–0.98) |
0.91 (0.80–0.97) |
NPV |
0.75 (0.43–0.93) |
0.56 (0.23–0.85) |
0.55 (0.25–0.82) |
Abbreviation: GGO, ground-glass opacity; NPV, negative predictive value; PPV, positive predictive value.
Comment 20.
The tile of Supplementary Table S1 must be provided, explaining what is presented in that table.
Response:
Thank you for your feedback. We have added the title "The Five Quantitative Epigenetic Features" for Supplementary Table S1, clearly explaining its content and relevance to the study.
Comments on the Quality of English Language
Comment 21.
Line 58: Type of a "resistance" must be specified.
Response:
Thank you for your valuable feedback. We have now specified the type of resistance as 'acquired or primary resistance' in the revised manuscript. We appreciate your careful review and suggestions.
“Consequently, while close monitoring is the primary strategy, larger tumors may develop acquired or primary resistance or metastasize during observation.”(page2, lines59)
Comment 22.
Line 64: Proper would be "microRNAs, circular RNAs".
Response:
Thank you for your careful review. We have revised the terminology from 'micro-RNA, circular RNA' to 'microRNAs, circular RNAs' in the revised manuscript as suggested.
“tumor cells release various biomolecules such as cell-free DNA (cfDNA), circulating tumor DNA (ctDNA), exosomes, micro-RNA, circular RNA microRNAs, circular RNAs, circulating tumor cells (CTCs)”(page2, lines63-65)
Comment 23.
Line 99: It is unclear what means "chest blood".
Response:
Thank you for pointing this out. 'Chest' was a typographical error. We have revised the sentence for clarity.
“All chest blood samples were prospectively collected before surgery, either during the initial nodule check or during the admission period.”(page3, lines111-112)
Comment 24.
Line 119: Abbreviation "%CV" must be explained
Response:
The abbreviation must be changed to “ percent coefficient of variation (%CV)”.
If further clarification is needed, the percent coefficient of variation (%CV) is a statistical measure of relative variability. It is calculated as the ratio of the standard deviation to the mean, expressed as a percentage. This metric is commonly used to assess and compare the variability of data across different datasets or measurement systems.
“All chest blood samples were prospectively collected before surgery, either during the initial nodule check or during the admission period.”(page3, lines111-112)
Comment 25.
Line 153: Abbreviation "EB" should be explained where it was first mentioned.
Response:
Thank you for your detailed review. We had previously introduced this in the Introduction: “We aimed to develop an epigenetic biomarker (EB) model.” Nonetheless, we appreciate your careful check.
Comment 26.
Figure 1 legend: Abbreviations "MN" and "BN" must be explained.
Response:
Thank you for your valuable suggestion. We completely agree. 'MN' refers to malignant nodule, and 'BN' refers to benign nodule. We have revised the figure legend accordingly.
“ Figure 1. Flowchart of participant enrollment and model development. A total of 806 patients with nodules >5 mm were enrolled. The Epigenetic Biomarker (EB) model was trained on 483 samples, validated on 121 samples, and independently tested on 202 samples. The model was built based on the training dataset, with the cutoff determined using the validation dataset, and its performance evaluated on the test dataset. MN refers to malignant nodules, and BN refers to benign nodules. ”(page5, lines177-178)
Comment 27.
'Discussion': Since H3.1 and H3K27Me3 levels were presented (measured) as concentration (ng/mL), statements like "H3K27me3 overexpression" and "H3K27me3 expression" are false, since those are not genes or proteins.
Response:
Thank you very much for your insightful comment. We completely agree that the wording may be misleading and could cause confusion. To clarify, the terms 'H3K27me3 overexpression' and 'H3K27me3 expression' do not accurately describe the measured concentrations. Instead, we are referring to an increase or decrease in the detected level of nucleosomes. We have revised the wording accordingly to ensure clarity.
“ Regarding the prognostic effect of H3K27me3 in various human cancers, increase level of H3K27me3 overexpression is linked to a more malignant behavior and worse prognosis in patients with prostate 33, esophageal 34, nasopharyngeal 35, and hepatocellular 36 carcinoma. Conversely, in breast, ovarian, and pancreatic cancers 31, and in renal cell carcinoma 37, a decrease in the H3K27me3 reduced H3K27me3 levels expression is associated with a worse prognosis.”(page11, lines365-370)
Reviewer 4 Report
Comments and Suggestions for Authors
The manuscript presented describes a very interesting and important tool for discriminating benign from malignant lung cancer noduli.
Previous publications focused with the same technology on late stage lung cancer, including not only histone modifications but as well correlation to cf/ctDNA mutation profile resulting in an improvement of diagnosis when combining these two detection methods.
So my first question would be whether there was any cf/ctDNA measured in the experimental setup in plasma and if the mutational status of the noduli was known after excision. If so, this information would be very interesting, in the sense that from early stage lung cancer it is very hard to get any ctDNA at all. Further question would then be, whether isolation of the ctDNA enriched by specific histone modification would increase the possibility to assess the mutation profile.
To understand the paper better it would be valuable to have insight about the NuQ results. What was the ng/ml measured from the H3.1 and the H327Me3 assays? What about the levels of the other three test markers (Table S1) which were discarded for the final model. Did the levels correlate with nodule size or nodule number? The latter point was not discussed at all in the paper: were there any patients with multiple noduli?
Interesting would be as well to compare the ng/ml levels between the patients with benign diagnosis due to disappearance of the noduli, although these would then give no further power in developing the model since for those no tissue analysis results were available.
The fine tuning of the model is not very clearly outlined. What exactely was fine tuned? Which parameters?
Finally a minor question to the methods: why did you not do a two-step isolation of the plasma with a high-spin centrifugation? As far as I can see from the NuQ method, nucleosomes are maintained in the plasma after tow-step centrifugation.
Author Response
Response to Reviewer 4:
Comment 1.
So my first question would be whether there was any cf/ctDNA measured in the experimental setup in plasma and if the mutational status of the noduli was known after excision.
If so, this information would be very interesting, in the sense that from early stage lung cancer it is very hard to get any ctDNA at all.
Further question would then be, whether isolation of the ctDNA enriched by specific histone modification would increase the possibility to assess the mutation profile.
Response:
Thank you for your insightful question. We agree that analyzing cf/ctDNA in plasma and assessing the mutational status of excised nodules could provide valuable information. Our team even attempted to integrate proteomic analysis. However, due to Institutional Review Board (IRB) regulations, the volume of blood samples collected from patients was limited, making it insufficient for additional analyses such as ctDNA or proteomic profiling.
We also acknowledge that detecting ctDNA in early-stage lung cancer is particularly challenging, which is one of the reasons we aimed to develop a new method. We believe that enriching ctDNA through specific histone modifications could potentially enhance the ability to assess the mutation profile, and this would be an interesting direction for future research, especially in the malignant group.’
Thank you for your kind suggestion. We have revised the 'Introduction' section accordingly to emphasize this point.
“PTMs regulate chromatin-mediated gene expression, affecting processes such as in-flammation, cell cycle, apoptosis, and tumor suppression in lung cancer. Nucleosomes are stable circulating nucleoprotein complexes carrying cfDNA and ctDNA. Unlike DNA analysis, which requires additional preparation and sequencing or PCR, nucleosomes can be directly quantified from plasma via immunoassay. Their measurement is fast, automatable, and suitable for clinical practice. Histone variants and modifications have shown prognostic significance in various cancers. However, their potential as biomarkers for lung nodule differentiation remains underexplored.”(page2, line79-86)
Comment 2.
To understand the paper better it would be valuable to have insight about the NuQ results. What was the ng/ml measured from the H3.1 and the H327Me3 assays?
What about the levels of the other three test markers (Table S1) which were discarded for the final model. Did the levels correlate with nodule size or nodule number? The latter point was not discussed at all in the paper: were there any patients with multiple noduli?
Response:
We appreciate the reviewer’s insightful comments. To clarify, in our previous study, NuQ levels were associated with both nodule size and disease stage. Patients with multiple nodules were not excluded; rather, the largest nodule was selected as the representative lesion for measurement and further analysis. Additionally, this test and current model are particularly valuable as it provides adequate diagnostic accuracy even for tumors as small as 5–10 mm.
We acknowledge the significance of this aspect and have now included the relevant results in the supplementary file for further reference.
Supplementary Table 2.
Summary of Epigenetic Marker Levels in Blood
|
|
Average (ng/mL) |
Median (ng/mL) |
Standard Deviation |
Min |
Max |
H3.1 |
nonCancer |
26.1 |
16.4 |
54.2 |
2.5 |
509.5 |
Cancer |
34.0 |
26.1 |
39.3 |
1.9 |
518.0 |
|
H3K27Me3 |
nonCancer |
16.6 |
11.7 |
25.1 |
3.8 |
261.7 |
Cancer |
17.8 |
14.6 |
16.3 |
2.7 |
252.3 |
|
H3K9Ac |
nonCancer |
4.9 |
4.3 |
4.5 |
1.9 |
57.4 |
H3K9Ac |
Cancer |
4.5 |
4.3 |
1.5 |
0.9 |
27.4 |
H3K9Me3 |
nonCancer |
17.4 |
13.1 |
22.5 |
4.8 |
246.0 |
H3K9Me3 |
Cancer |
17.8 |
15.1 |
12.5 |
4.3 |
187.0 |
H3K36Me3 |
nonCancer |
9.7 |
7.5 |
18.1 |
4.1 |
227.3 |
H3K36Me3 |
Cancer |
9.5 |
8.2 |
6.6 |
4.0 |
88.5 |
Comment 3.
Interesting would be as well to compare the ng/ml levels between the patients with benign diagnosis due to disappearance of the noduli, although these would then give no further power in developing the model since for those no tissue analysis results were available.
Response:
We appreciate the reviewer’s suggestion. While comparing ng/mL levels between patients with benign diagnoses due to nodule disappearance would be interesting, the absence of tissue analysis limits the potential for model development. Therefore, we have focused on patients with tissue-confirmed diagnoses as the primary study population for analysis.
Comment 4.
The fine tuning of the model is not very clearly outlined. What exactely was fine tuned? Which parameters?
Response:
We appreciate the reviewer’s inquiry. The details of the fine-tuning process are discussed in Section 3.2: "The EB model and lung cancer diagnostic accuracy."
To develop the EB model, we systematically screened multiple combinations of five quantitative epigenetic features (Table S1, Table S2) obtained from blood tests during the pre-training model tuning phase. For feature selection, we evaluated the relationship between the AUC values and the number of primary features. Our analysis showed that the AUC plateaued when the number of primary features reached two, indicating that adding more features did not significantly improve model performance.
Based on this observation, we identified two independent predictors of malignancy using multivariate logistic regression analysis. The final training dataset incorporated Nu.Q® H3.1 and Nu.Q® H3K27Me3 as the primary predictive markers.
Comment 5.
Why did you not do a two-step isolation of the plasma with a high-spin centrifugation? As far as I can see from the NuQ method, nucleosomes are maintained in the plasma after tow-step centrifugation.
Response:
We appreciate the reviewer’s question. Unlike cell-free DNA (cfDNA) or circulating tumor DNA (ctDNA) analysis, circulating nucleosome analysis does not require an extraction step, as nucleosomes can be directly measured from plasma samples.
Per Nu.Q® instructions for use, K2EDTA-plasma samples must undergo centrifugation (either single or double) within 4 hours of blood collection. Additionally, the first step of the Nu.Q® immunoassay includes a high-speed spin centrifugation to further clarify the plasma.
To address this point, we have included this clarification in the Materials and Methods section.
“ Briefly, plasma samples were centrifuged at high-speed during 2min and 50 µL of K2-EDTA plasma was incubated with acridinium ester labeled anti-nucleosome detection antibody.”(page3, lines124-126)

Round 2
Reviewer 2 Report
Comments and Suggestions for Authors
The manuscript has been improved and in my opinion it is suitable for publication.
Reviewer 3 Report
Comments and Suggestions for Authors
Authors have satisfactorily responded to my concerns and accordingly further improved quality of this manuscript.
Reviewer 4 Report
Comments and Suggestions for Authors
Dear authors
thanks for clarification and further added details.